# Dynamic Computed Tomography Angiography as Imaging Method for Endoleak Classification after Endovascular Aneurysm Repair: A Case Series and Systematic Review of the Literature

**DOI:** 10.3390/diagnostics13050829

**Published:** 2023-02-21

**Authors:** Gert Jan Boer, Ludo A. H. van Engen, Lievay van Dam, Koen M. van de Luijtgaarden, Reinoud P. H. Bokkers, Jean-Paul P. M. de Vries, Bram Fioole

**Affiliations:** 1Department of Vascular Surgery, Maasstad Hospital, 3007 AC Rotterdam, The Netherlands; 2Department of Radiology, Maasstad Hospital, 3007 AC Rotterdam, The Netherlands; 3Department of Radiology, Medical Imaging Center, University Medical Center Groningen, University of Groningen, 9713 GZ Groningen, The Netherlands; 4Department of Vascular Surgery, University Medical Center Groningen, University of Groningen, 9713 GZ Groningen, The Netherlands

**Keywords:** abdominal aortic aneurysm, EVAR, endoleak, computed tomography angiography, dynamic, time-resolved, time attenuation curve

## Abstract

Introduction: This study evaluated our experience with dynamic computed tomography angiography (dCTA) as a diagnostic tool after endovascular aortic aneurysm repair (EVAR) with respect to the endoleak classification and the available literature. Methods: We reviewed all patients who underwent dCTA because of suspected endoleaks after the EVAR and classified the endoleaks in these patients based on standard CTA (sCTA) and dCTA. We systematically reviewed all available publications that investigated the diagnostic accuracy of dCTA compared with other imaging techniques. Results: In our single-center series, 16 dCTAs were performed in 16 patients. In 11 patients, the undefined endoleaks that appeared on sCTA scans were successfully classified using dCTA. In three patients with a type II endoleak and aneurysm sac growth, inflow arteries were successfully identified using dCTA, and in two patients, aneurysm sac growth was observed without a visible endoleak on both sCTA and dCTA scans. The dCTA revealed four occult endoleaks, all of which were type II endoleaks. The systematic review identified six series comparing dCTA with other imaging methods. All articles reported an excellent outcome regarding the endoleak classification. In published dCTA protocols, the number and timing of phases varied greatly, affecting radiation exposure. Time attenuation curves of the current series show that some phases do not contribute to the endoleak classification and that the use of a test bolus improves the timing of the dCTA. Conclusions: The dCTA is a valuable additional tool that can identify and classify endoleaks more accurately than the sCTA. Published dCTA protocols vary greatly and should be optimized to decrease radiation exposure as long as accuracy can be maintained. The use of a test bolus to improve the timing of the dCTA is recommended, but the optimal number of scanning phases is yet to be determined.

## 1. Introduction

An endoleak is a common, long-term complication after the endovascular aneurysm repair (EVAR) that may cause aneurysm sac expansion and eventually lead to rupture [1]. Type I and III endoleaks need urgent surgical repair, whereas a type II endoleak without aneurysm sac expansion is often considered benign and can be treated conservatively in most patients [2,3].

In a routine follow-up of patients with an EVAR, duplex ultrasound (DUS) and computed tomography angiography (CTA) are the most frequently used imaging methods. A meta-analysis comparing CTA with DUS and contrast-enhanced duplex ultrasound (CE-DUS) found that the pooled sensitivity and specificity for detecting endoleaks with DUS were 0.77 and 0.94, respectively. CE-DUS was significantly more accurate, with a pooled sensitivity of 0.98 and specificity of 0.88 in endoleak detection [4]. CTA is, however, still considered the gold standard for classifying the endoleak type because this is more difficult on DUS. To identify endoleaks, a delayed phase is recommended to rule out flow in the aneurysm sac and to diagnose slow-flow endoleaks [2]. Nevertheless, the identification and classification of endoleaks on CTA may be challenging due to artifacts, the timing of the phases of the CTA, and a combination of multiple endoleaks and calcifications.

To more reliably detect the location and type of endoleaks, dynamic CTA (dCTA) has been recently introduced. Several studies evaluating the results of dCTA for postoperative endoleak detection and classification have been published in recent years [5,6,7,8,9,10]. The dCTA visualizes the entire aneurysm sac by a scan in the arterial phase and multiple scans thereafter, creating a dynamic overview of the endoleak.

The aim of this study was to evaluate our experience with dCTA after the EVAR as an additional diagnostic tool. We also conducted a systematic review and evaluation of the currently available dCTA protocols in the literature and their results.

## 2. Materials and Methods

All consecutive patients who underwent dCTA in addition to standard CTA (sCTA) after EVAR, thoracic EVAR, fenestrated EVAR, or branched EVAR between January 2020 and June 2022 in the Maasstad Hospital were retrospectively evaluated. dCTA has been available in our hospital since January 2020. The indications for dCTA, such as an undefined endoleak or aneurysm sac growth without apparent cause, were discussed in multidisciplinary meetings. Patient characteristics were retrospectively extracted from their medical files. The Medical Research Ethics Committees United and the Hospital Institutional Review Board approved this study. All patients provided written informed consent.

### 2.1. Imaging Protocol for Surveillance

The standard imaging protocol for postoperative surveillance was a CTA at 6 weeks and 1 year after surgery. Thereafter, patients received a yearly DUS or CTA on the indication. Indications for the continuation of CTA during a follow-up were an endoleak in combination with a stable aneurysm sac or growth of the aneurysmal sac. The indications for a dCTA were an aneurysm sac growth without a detectable endoleak on sCTA scans or the absence of aneurysm sac regression combined with an undefined endoleak on sCTA scans.

The first postoperative sCTA after 6 weeks consisted of non-contrast and delayed arterial phases. Images were acquired after 20 s, with a correction based on the time-to-peak of the test bolus in the aorta proximal of the stent. The test bolus contained 10 mL of a contrast agent, and the delayed arterial phase contained 75 mL of a contrast agent, with a contrast flow of 4 mL/s. The sCTA at the 1-year follow-up only consisted of a delayed arterial phase, and the non-contrast series of the first postoperative CTA was used to compare calcifications and stent material. The scan range was determined from the topogram. The sCTA tube voltage and tube current were determined by an automatic exposure device (AEC), CareDose 4D (Siemens Healthineers, Erlangen, Germany).

The dCTAs were acquired on a second-generation Dual Source SOMATOM Drive (Siemens Healthineers, Erlangen, Germany). For both sCTA and dCTA, Ultravist-300 (300 mg/mL; Bayer, Leverkusen, Germany) with jopromide (0.623 g/mL) was used as a contrast medium. Contrast infusion was managed with a CT injection system from MEDRAD Centargo (Bayer, Leverkusen, Germany). For both CTA methods, the contrast was administered through an 18-gauge infusion needle. No test bolus was administered during dCTA, but a standard delay of 8 s was used after 100 mL of contrast agent with a flow of 5 mL/s. The dynamic phase was performed using bidirectional table movement (shuttle mode) for longitudinal coverage. The dynamic phase acquisition image parameters were based on the automatic settings: 80 kV, 150 mAs, 32-mm × 1.2-mm detector configuration, 284-mm 4D range; number of scans, 16; rotation, 0.28 s; and total examination time, 41.2 s. Afterward, the venous phase was performed with a 128-mm × 0.6-mm detector configuration, 0.5 s, and a pitch of 0.6. Tube voltage and tube current were determined by the AEC. This protocol is based on the standard Siemens protocol.

### 2.2. Dose Length Product

The dose length product (DLP) of each scan was automatically reported with subdivision in scan phases, respectively, scout, test bolus, and delayed arterial or dynamic and venous phase. DLP was given in mGy × cm. No conversion was made to mSv.

### 2.3. Endoleak Classification

The types of endoleaks were qualitatively evaluated by two observers. For this, multiplanar images were reviewed using a standard workstation with JiveX Diagnostic Advanced (Visus Health IT, Bochum, Germany), and the dynamic images were analyzed with syngo.via, software version VB40 (Siemen Healthineers, Erlangen, Germany). The maximum diameter of the aneurysm sac was measured in double oblique reconstruction. The presence and the type of endoleaks were documented by both observers. In the case of a type II endoleak, the arteries involved were documented. If there was no agreement, a third observer (B.F.) made the final discission.

To differentiate between the times of contrast arrival within individuals, quantitative analysis was performed. Regions of interest (ROI) were drawn in the endoprosthesis, with the addition of endoleaks and arteries of interest. The ROI was manually drawn with the exclusion of aneurysm calcification and stent material (Figure 1). Time attenuation curves were given with time-to-peak in seconds and maximum peak in Hounsfield units. The maximum peak of the endoleak was divided by the maximum peak of the endoprosthesis.

## 3. Systematic Review

### 3.1. Protocol and Eligibility Criteria

This systematic review was conducted in accordance with the Preferred Reporting Items for Systematic Reviews and Meta-Analysis (PRISMA) statements [11]. Included were all studies that compared dCTA with other imaging methods for endoleak identification and classification after the EVAR, fenestrated EVAR, branched EVAR, and thoracic EVAR. Comparative imaging methods included CE-DUS, sCTA, and/or DSA. Publications were translated if they were not published in English or Dutch. Publications for which full texts were not available and studies that included less than 10 patients were excluded.

### 3.2. Search Strategy

An extensive search on PubMed, Embase, and Google Scholar was performed in June 2022. Search terms included EVAR, branched EVAR, fenestrated EVAR, thoracic EVAR, endoleak, dynamic CT angiography, dynamic CTA, time-resolved CTA, and time-resolved CT angiography. Reference lists were checked for additional publications. Because the first study was published in 2012, only publications thereafter were included. A flowchart of the selection process is shown in Figure 2.

### 3.3. Study Selection, Data Extraction, and Quality Assessment

Two observers (G.J.B. and L.E.) individually assessed all articles. If no agreement was reached, a third observer (B.F.) made the final discission. First, all titles of the search and then the abstracts of the remaining studies were reviewed. Finally, all remaining articles were read and included if they met the inclusion criteria. Information was extracted by G.J.B. and L.E. independently. The extracted variables were author, year of publication, number of patients, type of paper, type of imaging method compared with dCTA, CT scanner type, scan duration, number of phases, amount of contrast, use of a test bolus, tube voltage, tube current-time product, scanning range, and DLP. Study quality and risk of bias were assessed using QUADAS-2, a tool for the quality assessment of diagnostic accuracy studies [12].

### 3.4. Statistics

Statistical analysis was performed using RStudio 3.6.1 software (RStudio, PBC, Boston, MA, USA). Variables are presented as mean ± SD if normally distributed or as median (quartile 1–quartile 3) if not normally distributed.

## 4. Results

A total of 16 dCTAs were performed on 16 patients who already underwent the sCTA. Patients’ characteristics are presented in Table 1. The mean body mass index was 27.8 ± 3.7 kg/m^2^. The indication for dCTA in 11 patients was an undefined endoleak where a type I or III could not be excluded. In three patients, aneurysm sac growth and a type II endoleak were seen, with uncertainty about the inflow artery.

In all patients with undefined endoleaks on sCTA scans, endoleak types were identified with dCTA. A type Ia endoleak was identified in one patient, a type II endoleak in eight patients, and a type IIIb endoleak in two patients. In the three patients with type II endoleaks and aneurysm sac growth, inflow arteries were successfully identified on dCTA scans, and the type II endoleaks were confirmed. In the two patients with aneurysm sac growth without a visible endoleak, one type IIIb endoleak was detected. The dCTA also revealed four occult endoleaks, all of which were type II endoleaks.

### 4.1. Time Attenuation Curves

There was a wide heterogeneity in the arrival time of contrast within the endograft in the patients. The time-to-peak attenuation of the endoprosthesis lumen showed a wide range from 18.8 to 33.4 s. Time-to-peak attenuation of the endoleak varied from 23.7 to 38.5 s. In seven patients, the peak attenuation of the endoprosthesis lumen or endoleak was not yet reached at the end of the maximum delay of 41.2 s. The mean delay between the peak of the endoprosthesis lumen and type I or III endoleaks was 4.4 ± 1.0 s, whereas the mean delay between the peak endoprosthesis lumen and a type II endoleak was 7.8 ± 1.8 s (Figure 1 and Figure 3). Compared with the endoprosthesis lumen, the mean peak attenuation of type I and III endoleaks was 89.2% ± 2.9%, and the peak attenuation of the type II endoleak was 70.1% ± 32.3%. Differences in peak attenuation were only calculated if the maximum peak was achieved (Table 2). The time attenuation curves of patient 15 helped to determine the inflow and outflow vessel of the endoleak. In this case, infolding of the proximal sealing stent of the endoprosthesis caused a type Ia endoleak, and the inferior mesenteric artery was the outflow vessel of the endoleak (Figure 4).

### 4.2. Systematic Review

We identified 261 studies. Six studies comparing dCTA with CE-DUS, sCTA, or DSA were included, comprising four retrospective studies and two prospective studies. These studies included between 12 and 69 patients. A detailed overview of the study characteristics is provided in Table 3. Quality assessment is provided in Table 4.

#### 4.2.1. Acquisition Protocols

Four series used a test bolus to determine the contrast medium transit time. Three series measured the arrival of contrast at the top of the endoprosthesis, and one series measured the contrast arrival in the pulmonary artery. The series without a test bolus used a standard delay of 13 s and 2 s, respectively [8]. In the protocol reported by Apfaltrer et al. [8], 1 scan per 4 s for 12 acquisitions, followed by 3 additional acquisitions with 1 scan per 10 s, was performed. The total amount of contrast that was administered varied from 60 to 160 mL, but in most patients less than 90 mL of contrast was used.

Radiation exposure varied from 505 to 11,069 mGy*cm. Five publications reported less than 1100 mGy*cm of radiation. In the current series, the mean DLP of dCTA was 1815 ± 215 mGy*cm. If the venous phase was excluded from de dCTA, the mean DLP was reduced to 1524 ± 139 mGy*cm (Table 3). In patient 2, the automatic settings of the tube current were elevated from 150 mAs to 180 mAs due to his body mass index, which was more than 35 kg/m^2^. In this patient, the DLP of the dCTA increased to 3165 mGy*cm and without the venous phase to 2638 mGy*cm. An overview of the dCTA protocols is provided in Table 3.

#### 4.2.2. dCTA versus CE-DUS

One study reported a comparison with CE-DUS. In a prospective study by Sommer et al. [5], 48 patients with known or suspected endoleaks and patients with high risk for endoleaks because of the challenging anatomy were examined following the EVAR using both dCTA and CE-DUS, with CE-DUS as a reference. They detected 19 endoleaks using dCTA and 18 endoleaks using CE-DUS. The dCTA detected two endoleaks that were not found with CE-DUS. One type Ib endoleak detected with CE-DUS was not identified on dCTA. Overall sensitivity and specificity for dCTA were 94% and 93%, respectively [4].

#### 4.2.3. dCTA versus sCTA

Three studies, with a total of 101 patients, reported outcomes of dCTA compared with sCTA [6,8,10]. Tarulli et al. [6] reported 13 patients who underwent additional dCTA because of an undefined endoleak after fenestrated or branched EVAR on sCTA scans. They found an overall sensitivity of 100% for endoleak detection and a specificity of 87.5% for the endoleak classification in dCTA, when sCTA was used as a reference. Apfaltrer et al. [8] found one additional endoleak on dCTA in 19 consecutive patients who underwent sCTA. Finally, Waldeck et al. [10] identified 11 endoleaks in 10 of 50 patients (20%) with sCTA, predominantly type I endoleaks (8× type I, 3× type II). However, dCTA identified 44 endoleaks in 26 of 69 other patients (37.7%) and the detected endoleaks were more evenly distributed (20× type I, 19× type II, 4× type III) reflecting a more accurate and realistic detection and classification of suspected endoleaks.

#### 4.2.4. dCTA versus DSA

Two studies, with a total of 36 patients, reported a comparison between dCTA and DSA [7,9]. Hou et al. [7] included 12 patients with undefined endoleaks on sCTA scans. They classified four type I, seven type II, and one type III endoleaks on the dCTA, all of which were confirmed with DSA. Berczeli et al. [9] compared 24 patients who underwent dCTA with DSA, which was used as the gold standard. They identified four type I, sixteen type II, and two type III endoleaks in twenty-two patients, and no endoleaks in the remaining two patients. The results in 23 of 24 patients matched the DSA findings. One type III endoleak was only detected on the dCTA scan.

## 5. Discussion

The current series shows that dCTA is a valuable imaging method to classify undefined endoleaks on sCTA scans (Figure 5). In our patients, all undefined endoleaks on sCTA scans were successfully classified by the dCTA protocol. These findings are in line with other reports [7,13]. The current series indicates that type II endoleaks were occult in three of sixteen patients. In the current literature, dCTA has shown excellent sensitivity and specificity in endoleak detection and/or classification compared with sCTA or DSA. CE-DUS appears to be an equivalent excellent imaging modality; however, few studies are available to substantiate this conjecture.

Our systematic review shows that type II endoleaks may be missed or misclassified on sCTA scans due to the delayed arterial phase that is used in the sCTA protocol. Lehmkul et al. [14] found that most type II endoleaks were identified in phase 6 of the dCTA, which was 27 s after contrast administration. Sommer et al. [15] showed that a type I endoleak is identified nearly simultaneously with peak contrast enhancement of the aortic lumen, but a type II endoleak appeared 8.3 s later. A study by Koike et al. reported the same results. [16] This finding supports that in a triphasic CTA protocol, type II endoleaks still may be occult because they may arise after the arterial phase and extinguish before the venous phase.

In patients with type II endoleak, dCTA also may be useful in identifying the arteries that supply the flow to the aortic sac. To treat a type II endoleak, it is important to differentiate between the endoleak originating from the inferior mesenteric artery and the endoleak originating from a lumbar or intercostal artery. In our series, when patients had aneurysm sac growth due to a type II endoleak, dCTA successfully identified the inflow and outflow arteries related to the endoleak. The preoperative plan based on dCTA ensured the significant accuracy of the intervention. Berczeli et al. [9] also demonstrated that the dCTA is accurate in identifying inflow arteries. In type II endoleaks, they detected more inflow arteries on dCTA than on DSA imaging (33 vs. 21 arteries). Hou et al. [13] showed that preoperative dCTA resulted in fewer angiograms to assess the inflow arteries of the endoleak (1 vs. 6). This resulted in less contrast use and reduced radiation dosage.

### 5.1. Radiation Exposure

Radiation exposure was significantly higher in our series compared with all other reported dCTA studies (Table 3), except for one. In comparison with the other protocols, this is mainly due to a higher tube current and a higher number of phases. The venous phase after the dCTA had no additional value and will be omitted to reduce radiation exposure. A better timing or reduction of the phases in the dCTA protocol may further reduce radiation exposure.

### 5.2. Timing of the Phases in the dCTA Protocol

The current guideline recommends sCTA for postoperative surveillance after the EVAR, and a delayed phase is advocated for ruling out endoleaks [2]. However, no recommendation is made with respect to the timing of the delayed phase. In the current series, the peak attenuation of type II endoleaks was 7.8 s after the peak of the endoprosthesis lumen compared with 4.4 s in type I or type III endoleaks, which is in line with other reports [14,15]. This suggests that in a single delayed phase, some endoleaks might be missed.

In our dCTA protocol, 16 phases were scanned. The time attenuation curves showed that the peak of the endoprosthesis lumen was observed after 18.8 s, and the peak of the endoleak was observed 23.7 s after the start of scanning. In almost half of the endoleaks, the maximum peak was not reached in the last phase. Moreover, as shown in Figure 3, a number of phases did not contribute to the visibility of endoleaks. This insufficient timing might be due to the standard delay of 8 s and may be overcome with the use of a test bolus.

When we reviewed our own dCTA protocol and compared it with other published protocols, it was clear that there are improvements to be made. A test bolus must be administered to eliminate most non-contributing phases before the start of the peak attenuation of the endoprosthesis lumen. Because the peak attenuations of type I/III and II endoleaks were, respectively, 4.4 and 7.8 s after the peak attenuation of the endoprosthesis lumen, 6 to 8 phases with an interval of 2 s seemed to be sufficient to identify and classify endoleaks.

Because this case series consists of only 16 patients, drawing firm conclusions from the results is not possible. In addition, the indication for dCTA was subjective, and the protocol was not yet properly adjusted. Although in practice the dCTA remains a very useful imaging method, its actual sensitivity and specificity remain unclear. The gold standard to which dCTA should be compared is still unknown. CE-DUS also might be a good alternative for endoleak detection, but further research is warranted to prove this. Finally, as shown by Waldeck et al. [10], the occult endoleaks that were missed on sCTA were most often type II endoleaks, which can be treated conservatively if no aneurysm sac is observed.

## 6. Conclusions

The dCTA is a valuable additional tool to the regular follow-up to classify unclassified endoleaks. CE-DUS also seems to be a good imaging method for endoleak detection and classification. The use of a test bolus to improve the timing of the dCTA is recommended, but the optimal number of scanning phases is yet to be determined. In our opinion, CE-DUS and sCTA should still be the first-choice imaging after the EVAR. However, if post-EVAR unclassified endoleaks or aneurysm sac growth without an endoleak are observed, an dCTA should be performed.

## Figures and Tables

**Figure 1 diagnostics-13-00829-f001:**
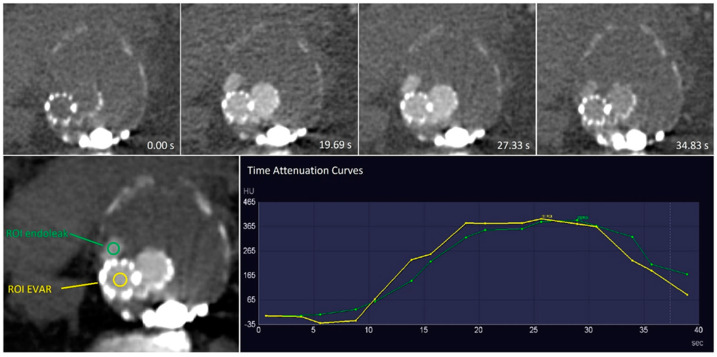
Example of the use of a region of interest (ROI) in a type IIIa endoleak. The images in the top row show different dynamic computed tomography angiography phases given in seconds. The ROI was used to determine the time attenuation curves of the endoprosthesis lumen and the endoleak by dynamic vessel evaluation combined with the time-to-peak of the endoprosthesis lumen and endoleak. Both curves were similar, and in combination with the location of the endoleak and the lack of component disconnection, this endoleak was defined as type IIIb.

**Figure 2 diagnostics-13-00829-f002:**
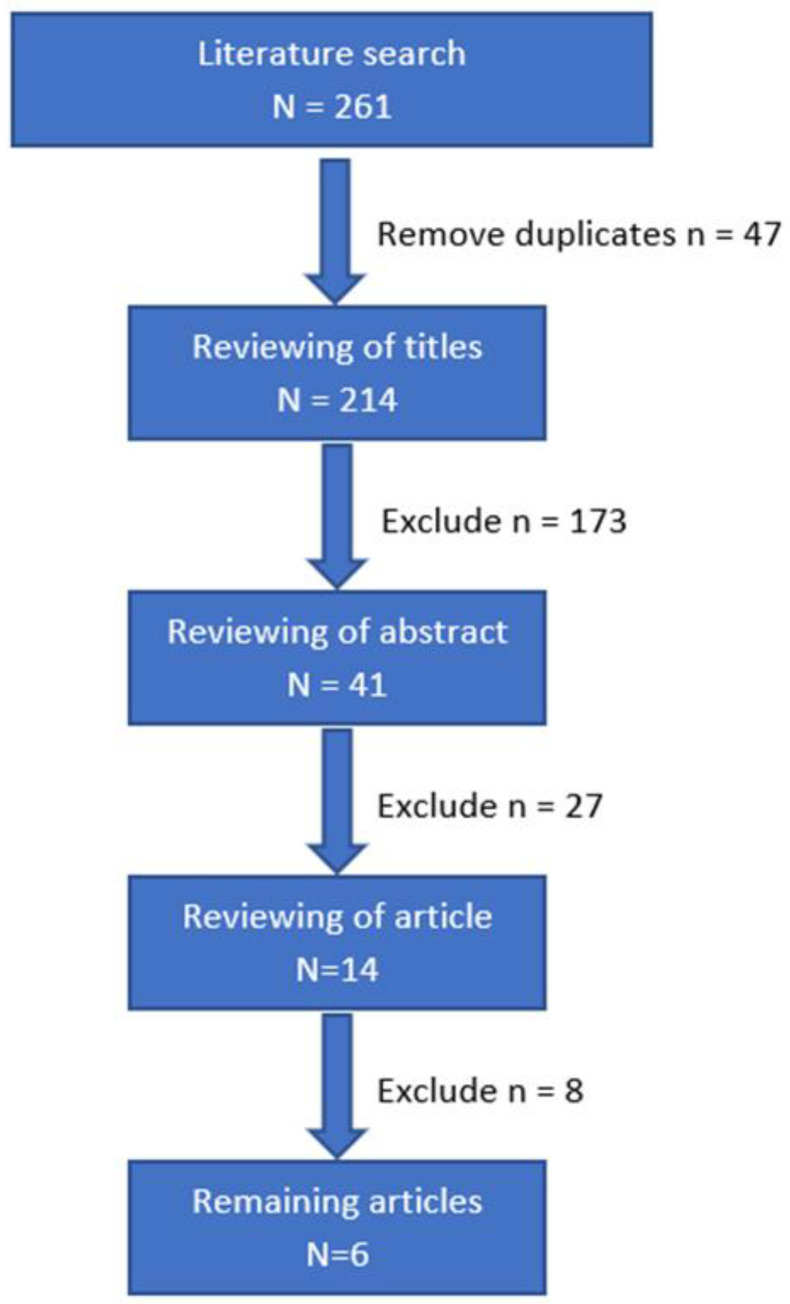
Flowchart of the selection process.

**Figure 3 diagnostics-13-00829-f003:**
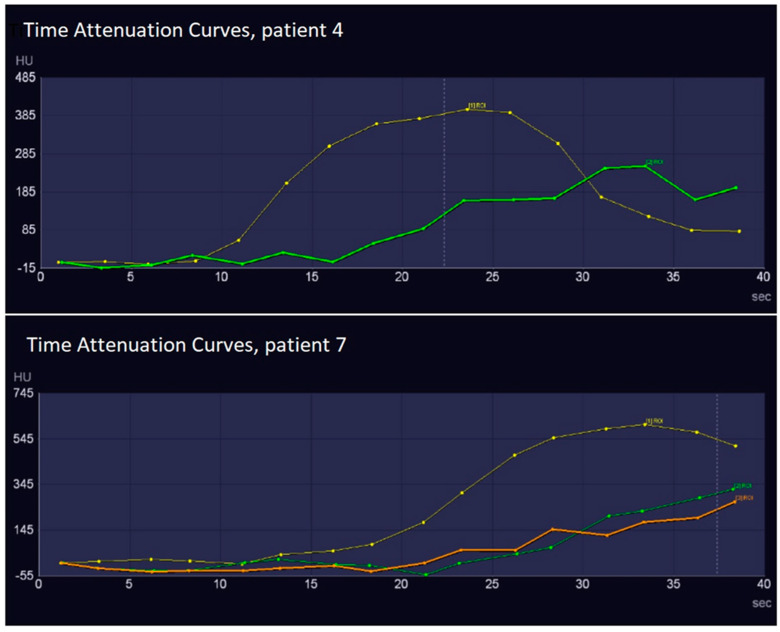
The time attenuation curves of the dynamic computed tomography angiography in two patients: patient 4 (type II endoleak) and patient 7 (2× type II endoleaks). Owing to the fixed delay of 8 s after the contrast bolus, the time-to-peak of the stent varied per patient as a result of the different cardiac outputs. In patient 7, it is unknown whether the maximum peak of the endoleak was reached.

**Figure 4 diagnostics-13-00829-f004:**
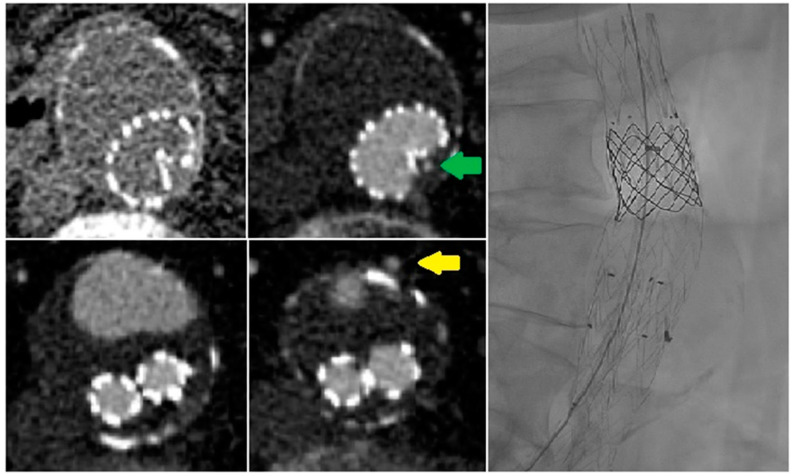
In patient 15, infolding of the posterior part of the proximal sealing stent of the endoprosthesis was observed in combination with a type Ia endoleak (green arrow). Because most of the endoleak was present in the anterior part of the aneurysmal sac in close relation to the inferior mesenteric artery (yellow arrow), there was uncertainty about the type of endoleak: type Ia, type II, or both. After dynamic computed tomography angiography, a type Ia endoleak was confirmed, and the patient was successfully treated with a large balloon-expandable stent re-expanding the proximal sealing stent of the endoprosthesis.

**Figure 5 diagnostics-13-00829-f005:**
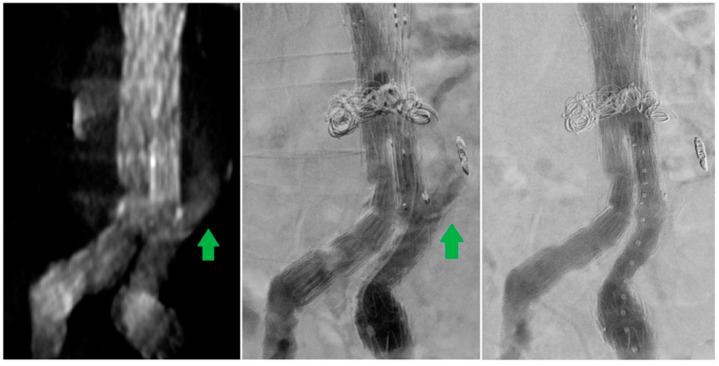
Patient 16 with a fenestrated endograft previously treated for a type II endoleak from the inferior mesenteric artery with coils. The standard computed tomography angiography showed a persistent endoleak with unknown origin. (not shown) The dynamic computed tomography angiography showed the origin from the left iliac branch (green arrow).

**Table 1 diagnostics-13-00829-t001:** Patients’ characteristics, sCTA and dCTA. sCTA = standard computed tomography angiography. dCTA = dynamic computed tomography angiography.

Patient No.	Age (Years)	Sex	Initial Diameter (mm)	Diameter Increase (mm)	Time after Initial Operation (months)	Endoleak Classification on sCTA	Indication for dCTA	Endoleak Classification on dCTA
1	78	Male	58	+1	18	1× type II, 1× undefined	Undefined EL (type Ia or II)	2× type II
2	77	Male	56	+20	39	1× undefined	Undefined EL (type Ia or II)	1× type II
3	88	Male	57	+10	129	None	AAA diameter increase without an endoleak	1× type IIIb
4	84	Male	59	+10	54	1× undefined	Undefined EL (type Ia or II)	1× type II
5	83	Female	57	+8	9	1× undefined	Undefined EL (type Ia or II)	1× type IIIb
6	87	Male	59	+7	81	1× type II	AAA diameter increase with unclear origin of type II EL	2× type II
7	74	Male	74	+8	85	1× type II	AAA diameter increase without an endoleak	2× type II
8	84	Male	59	+1	62	1× undefined	Undefined EL (type Ib or II)	3× type II
9	67	Male	38	+8	34	1× type II	AAA diameter increase with unclear origin of type II EL	1× type II
10	78	Male	57	None	15	1× undefined	Undefined EL (type Ia, II or III)	1× type II
11	66	Male	53	+3	3	1× undefined	Undefined EL (type II or III)	1× type II
12	89	Female	54	+12	93	None	AAA diameter increase without an endoleak	None
13	73	Male	62	+8	25	1× undefined	Undefined EL (type II or III)	1× type II
14	83	Male	56	+6	35	1× undefined	Undefined EL (type Ia or II)	1× type II
15	73	Female	54	None	8	1× undefined	Undefined EL (type Ia or II)	1× type Ia
16	66	Male	49	+2	74	1× undefined	Undefined EL (type II or III)	1× type IIIb

**Table 2 diagnostics-13-00829-t002:** Peak attenuation curves and time to peak. * curve is not finished yet.

Patient No.	Endoleak	Branch	Peak Stent (HU)	Peak Endoleak	Peak Endoleak/ Stent (%)	Time Peak Stent (s)	Time Peak Endoleak (s)
1	2× type II	AMI	348.8	332.4	95.3	20.6	28.9
		L1	348.8	333.6	95.6	20.6	28.6
2	1× type II	L1	370.9	304.1	82	28.5	33.3
3	1× type IIIb		459.9	425.8	92.6	25.6	28.9
4	1× type II	L1	401.2	252.1	62.8	23.6	33.4
5	1× type IIIc		NA	NA	NA	NA	NA
6	2× type II	AII	607.4	268.3	44.2	33.4	MAX *
		L1	607.4	323.6	53.3	33.4	MAX *
7	2× type II	L1	538.9	186.6	34.6	28.6	MAX *
		L2	538.9	171	31.7	28.6	MAX *
8	3× type II	AMI	496.1	350	70.6	MAX *	MAX *
		L1	496.1	341.7	68.9	MAX *	MAX *
		L2	496.1	138.8	28	MAX *	MAX *
9	1× type II	L1	549.9	232.6	42.3	23.5	33.5
10	1× type II	NA	NA	NA	NA	NA	NA
11	1× type II	NA	NA	NA	NA	NA	NA
12	None		NA	NA	NA	NA	NA
13	1× type II	L1	473.1	413	87.3	24.2	30.5
14	1× type II	L1	306.1	78.1	25.5	30.8	38.5
15	1× type Ia		466	408.6	87.7	18.8	23.7
16	1× type IIIb		473.1	413.4	87.4	28.4	33.5

**Table 3 diagnostics-13-00829-t003:** Overview of the series that reported on conventional imaging vs. dCTA. DSA = digital subtraction angiography, CE-DUS = contrast-enhanced duplex ultrasound, sCTA = standard computed tomography angiography, and dCTA = dynamic computed tomography angiography. * Dynamic CTA-protocol without additional venous phase. ** 350 mAs with 0.5 s per acquisition.

Author	Year	No.	Type of Study	dCTA vs.	CT Scanner Type	Scan Duration (s)	No. of Phases	Contrast (mL)	Test Bolus	Tube Voltage (kVp)	Tube Current-Time Product (mAs)	Range (cm)	DLP * (mGy*cm)
Sommer et al. [5]	2012	54	Prospective	CE-DUS	128-section Somatom Definition AS+ (Siemens)	30–60	12	60	No	80	120	27	952 ± 42
Hou et al. [7]	2019	12	Prospective	DSA	320-row detector (Aquilion One, Toshiba)	24–32	12–16	55–74	Yes	80	120	16	505–566
Apfaltrer et al. [8]	2020	19	Retrospective	sCTA	Third-generation, dual-source (Siemens)	NA	12	50	No	70	200	NA	1064–1065
Berczeli et al. [9]	2022	24	Retrospective	DSA	Third-generation, dual-source (Siemens)	39	10–12	70–90	Yes	84–110	150	NA	1038 ± 533
Tarulli et al. [6]	2022	13	Retrospective	sCTA	320-row detector (Aquilion One, Toshiba)	120	10–40	70–160	Yes	100	175 **	16	4724 (1108–11,069)
Waldeck et al. [10]	2022	69	Retrospective	sCTA	320-row detector (Aquilion One, Toshiba)	14	10	60–70	Yes	90	adaptive	36	856 ± 55
Current series	2022	16	Retrospective	sCTA and DSA	Second-generation, dual-source (Siemens)	41.2	16	100	No	80	151 ± 3.5	28.4	1524 ± 139

**Table 4 diagnostics-13-00829-t004:** Quality assessment of all articles.

Study	Risk of Bias	Applicability Concerns
	Patient Selection	Index Test	Reference Standard	Flow and Timing	Patient Selection	Patient Selection	Index Test
Sommer et al. [5]			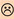		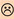		
Hou et al. [7]	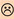		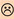				
Apfaltrer et al. [8]	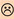		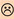				
Berczeli et al. [9]	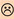		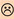				
Tarulli et al. [6]		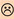	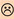	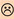			
Waldeck et al. [10]	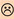		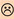	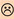	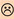		
 Low Risk 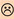 High Risk

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
