# Peer review of "Dynamic Computed Tomography Angiography as Imaging Method for Endoleak Classification after Endovascular Aneurysm Repair: A Case Series and Systematic Review of the Literature"

_diagnostics, 2023, doi:10.3390/diagnostics13050829_

Round 1

Reviewer 1 Report

I would like to congratulate the authors on their work! This is potentially significant research regarding the role of dCTA after EVAR as an additional diagnostic tool. The authors also conducted a systematic review and evaluated the currently available dCTA protocols in the literature and their results.

However, there are several aspects that must be addressed.

For a systematic review, there are too few references (only 15). I strongly advise the authors to improve the quality of the research by citing latest papers regarding the CTA and dCTA and outcome of patients with AAA. For example: https://doi.org/10.3390/ijerph192315961

https://doi.org/10.1148/radiol.13120197

https://doi.org/10.1016/j.jvir.2014.03.019

https://doi.org/10.1177/1526602801008005

            Overall, there are just minor comments regarding the English, but that do not influence the quality of the manuscript.

            Kind regards

Author Response

Response to Reviewer 1 Comments

We would like to thank the reviewer for his/her compliments about our manuscript and his/her efforts to review our manuscript.

Point 1: For a systematic review, there are too few references (only 15). I strongly advise the authors to improve the quality of the research by citing latest papers regarding the CTA and dCTA and outcome of patients with AAA. For example:

https://doi.org/10.3390/ijerph192315961

https://doi.org/10.1148/radiol.13120197

https://doi.org/10.1016/j.jvir.2014.03.019

https://doi.org/10.1177/1526602801008005

Comment 1: We agree with the reviewer that there are too few references. However, sinds dynamic CTA is a relative new imaging modality and has not been used often in vascular surgery, the number of articles that described dynamic CTA vs standard CTA or that described the use of dCTA in endoleak classification/detection is limited. We thank the reviewer for the examples. However, the first article that was proposed is not about dCTA, the second article is already used as a reference in our manuscript and the fourth is a non-existent doi. We absolutely do agree that the article by koike et al. is an excellent article and we have added this article to our references.

Reviewer 2 Report

Defining type 2 endoleaks in particular is difficult.  Could the authors comment on whether dCTA allowed a directed intervention for the type 2 endoleaks?  Did this ma out the anatomy enough to make a difference?  How does this approach compare to prone CTA for defining endoleaks?

Author Response

Response to Reviewer 2 Comments

At first, we would like to thank the reviewer for his/her comments on our manuscript. We appreciate the time spent to read our manuscript. We would like to answer the question the reviewer has.

Point 1. Defining type 2 endoleaks in particular is difficult.  Could the authors comment on whether dCTA allowed a directed intervention for the type 2 endoleaks?  Did this ma out the anatomy enough to make a difference?  How does this approach compare to prone CTA for defining endoleaks?

Response 1. We agree with the reviewer that defining type 2 endoleaks is difficult. dCTA as an imaging modality allowed us to prepare better for angiography, if needed. As we described on page 11, line 288-306, dCTA helped us differentiate between an endoleak originating from the inferior mesenteric artery or a lumbar or intercostal artery. In our series we were always able to identify the inflow and outflow arteries. On triphasic CTA it was not always possible to determine what type of endoleak it was and inflow or outflow arteries could not be identified.